

# New insights into the impact of wood vinegar on the growth and rhizosphere microorganisms of cherry radish (*Raphanus sativus* L.)

Shiguo Gu[1], Wei Zhu[1], Liying Ren[1], Binbin Sun[2], Yuying Ren[3], Yongkang Niu[1], Xiaokang Li[4] and Qingshan He[3]

[1] College of Civil and Architecture Engineering, Chuzhou University, Chuzhou, China
[2] Institute of Pollution Control and Environmental Health, School of Energy and Environmental Engineering, Hebei University of Technology, Tianjin, China
[3] School of Environment and Ecology, Jiangnan University, Wuxi, China
[4] School of Environmental and Material Engineering, Yantai University, Yantai, China

Corresponding authors
Binbin Sun, sunbinnaruto@126.com
Qingshan He, 287314371@163.com

## ABSTRACT

Understanding the impact of wood vinegar on the growth of cherry radish is indispensable for use in crop production and environmental safety. Our study explored the regulation of rhizosphere microbial abundance and activity by wood vinegar, as well as the relationship between microbial community and growth factors in-depth and systematically. Bacterial communities at the phylum and genus levels were significantly changed after wood vinegar treatment. Application of 200-fold diluted wood vinegar significantly boosted *Actinobacteriota* and *Firmicutes* abundances by 40.88% and 126.67%, respectively, while *Proteobacteria* was promoted in carbon-rich soil. Fungi positively responded to cherry radish root traits and were correlated with aboveground biomass and fruit production. The fungi that correlated with photosynthesis included *Albifimbria*, *Allomyces*, *Calcarisporiella*, *Clonostachys*, *Fusarium*, *Fusicolla*, *Knufia*, *Nigrospora*, *Paraconiothyrium*, *Preussia*, *Talaromyces*, and *Mortierellomycota*. Wood vinegar treatment significantly affected the composition and abundance of soil bacterial and fungal communities in cherry radish rhizosphere, while simultaneously enhancing photosynthetic efficiency (*e.g.*, Pn: 80.45% and Tr: 56.75%) and resulting in a 44.91% increase in crop yield. The promotion of cherry radish growth by wood vinegar may be attributed to the stimulation of soil microorganisms that degraded aromatic compounds and drove nitrogen cycling. This study provided novel insights into the significant promotion of cherry radish growth using wood vinegar diluted 200 times and identified potential microbial targets for agricultural applications.

## INTRODUCTION

Agricultural and forestry waste, which is approximately 2 billion tons generated annually in China (*Fan et al., 2020*), is an important biomass resource as a byproduct generated during the production and processing of agroforestry (*Ozturk et al., 2017*; *Sun et al., 2023*; *Yu et*

*al., 2024*). Wood vinegar obtained *via* condensing and separating the flue gas generated from biomass pyrolysis (*Hagner et al., 2013*; *Yin et al., 2024*), which has a wide range of applications in agriculture, environmental protection, and healthcare (*Grewal, Abbey & Gunupuru, 2018*; *Mahmud et al., 2016*; *Hu & Gholizadeh, 2020*; *Sarchami, Batta & Berruti, 2021*; *Feng et al., 2023*). *Mmojieje & Hornung (2015)* reported that wood vinegar from mixed wood biomass exhibited up to 90% mortality for red spider mite and green peach aphid. *Gao et al. (2020)* reported that wheat straw vinegar containing phenols, acetic acid, and cations significantly decreased the wheat fusarium head blight infection rate and deoxynivalenol content by 66% and 69%, respectively. Recently, research on the application of wood vinegar had mainly focused on their antibacterial properties and insecticidal effects in agriculture. Nevertheless, the sawdust vinegar from low temperature (<400 °C) could be used as liquid fertilizer to promote wheat seeds germination (*Shang et al., 2021*). These results almost focused on the biological response of wood vinegar to the target substance (plants, pests, and pathogens), but limited study concerned the wood vinegar effect on soil microbiome (*Jeong et al., 2015*), and further studies for the rhizosphere microbial behavior of wood vinegar are warranted.

Wood vinegar, as soil remediation agents, consisted of alcohols, esters, amines, pyridines, as well as trace elements such as potassium (K), phosphorus (P), calcium (Ca), manganese (Mn), and ferrum (Fe) (*Hou et al., 2018*; *Lu et al., 2020*). As reported by many previous studies, wood vinegar could increase soil fertility by increasing nutrient elements (*e.g.*, nitrogen (N), P, and K) (*Polthanee, Kumla & Simma, 2015*; *Sun et al., 2018*; *Mirsoleimani et al., 2023*), reducing $NH_3$ volatilization (*Win et al., 2009*), and dissolved organic molecules (*Lashari et al., 2013*; *Lashari et al., 2015*; *Fu et al., 2023*), resulting in improved production of crop. Wood vinegar gotten from biomass pyrolysis at low temperature (<150 °C) are mainly composed of acid compounds, which promote the length of wheat main roots to nearly 1.2 times (*Lu et al., 2019a*; *Lu et al., 2019b*). *Lashari et al. (2013)* reported that biochar poultry manure compost fortified with wood vinegar significantly increase wheat yield through wood vinegar leaching soluble salts and enhancing the effect uptake of P and K. Moreover, the dose of 3.0% mL hazelnut shell wood vinegar had positive effects on the number of bacteria and β-glucosidase enzyme activity in soil (*Koç et al., 2019*). In general, wood vinegar can improve crop yield and microbial activity, but most of the previous studies separately confirmed the effect of wood vinegar on soil microorganisms and plants (*Jeong et al., 2015*), and the coupling relationship was not introduced in results, especially the regulation of rhizosphere microorganisms through wood vinegar leading to crop growth. Therefore, the effects of wood vinegar composition on the response of soil microorganisms and the promotion of cherry radish growth were studied, and its applications were further determined.

In this study, we used wood vinegar produced from pyrolysis of rice straw. We deeply explored the influence of wood vinegar on the rhizosphere microbial community of cherry radish through soil culture experiments. In addition, the effects wood vinegar concentrations on the growth indicators, photosynthetic efficiency, and chlorophyll content of cherry radish were also explored. The objective is to explore the potential of wood vinegar in activating rhizosphere microorganisms, thereby enhancing photosynthetic

efficiency in crops and promoting the uptake of nutrients in the rhizosphere. These results could provide new insights into the resource utilization of agricultural waste, and important support for green agricultural development.

## MATERIALS AND METHODS

### Materials

The soil samples were collected from Ma'anshan in Anhui, classified as brown soil, and developed from quaternary loess parent material. The cultivated brown soil has a pH value of 6.5, 32.38 g/kg of organic matter, 15.12 g/kg of total nitrogen, 15.26 mg/kg of available phosphorus and 56.58 mg/kg available potassium. The ''Red Angel'' cherry radish used as the experiment crop and was purchased from Beijing Dongsheng Seed Industry Co., Ltd. 20.00 g of rice straw was weighed and then placed into a quartz tube inside a tube furnace. With a continuous nitrogen flow of 100 mL/min, the sample was gradually heated to a pyrolysis temperature of 450 °C at a rate of 10 °C/min and then held at that temperature for 120 min (*Hua et al., 2020*). The crude wood vinegar was collected from three separate pyrolysis experiments and gathered in an amber sample bottle. Following a 20-day storage period in a dark environment, the sample underwent a meticulous separation process to yield the wood vinegar.

### Experimental design

Cherry radish seeds were immersed in a 4% (v/v) sodium hypochlorite (NaClO) solution for 10 min to disinfect them, then rinsed with deionized water several times to remove residual disinfectants (*Lian et al., 2022*). Firstly, five seeded seeds were sown directly at a depth of 1 cm on the surface of bowl soil (3 kg). After 7 days of growth, only one plant per pot, which exhibited good and consistent growth, was retained. Cherry radish was cultivated in a greenhouse with a 12-h light cycle, day/night temperature of 25/20 °C, and 60% relative humidity (*Gu et al., 2022*). After 45 days of growth, the cherry radish was harvested and related indicators were measured in the soil. Each group of treatments was set with five replicates (five plants), and a total of five treatments were set in the experiment: the untreated wood vinegar was set as the control (CK), and the wood vinegar was diluted with deionized water by 400 (W400), 300 (W300), 200 (W200), 100 (W100), and 50 (W50) times, respectively. During the experiment, 500 mL of wood vinegar with different dilution ratios were poured into pots and bowls, and the same volume of water was poured into CK.

### Characterization of soil and wood vinegar

The soil pH is measured using the potential method (soil-water ratio 2.5:1). The organic matter of the soil was determined using potassium dichromate-sulfuric acid ($K_2Cr_2O_7$-$H_2SO_4$) for oxidation and ferrous sulfate ($FeSO_4$) titration (*Peng et al., 2016*). The soil sample is digested using copper sulfate ($CuSO_4$) and potassium sulfate ($K_2SO_4$), and then titrated with hydrochloric acid (HCl) to obtain total nitrogen (*Lenaerts et al., 2018*). The available phosphorus and potassium were determined by modified kelowna methods (*Qian, Schoenaru & Karamanos, 1994*).

The molecular composition of wood vinegar samples measured using an electrospray ionization (ESI) Fourier transform ion cyclotron resonance mass spectrometry (FT-ICR MS) (Bruker SolariX, Bruker, Germany) equipped with a 9.4 T superconducting magnet. Sample solutions were injected into the electrospray source at 180 µL/h with a syringe. The operating conditions for negative-ion formation be applied with a 4.0 kV spray shield voltage, 4.5 kV capillary column introduced voltage, and 320 V capillary column end voltage. The mass range was configured with 200–800 Mass to charge ratio (m/z). The samples were given 128 FT-ICR MS scans to improve signal-to-noise ratio and dynamic range. The detailed determinations of H/C, O/C, and double bond equivalents (DBE) of molecular compositions are presented in Text S1 (*Hua et al., 2020*).

## Measurements of plant growth

Cherry radish fruits were harvested after ripening, and the diameter of the fruit was measured using a ruler (centimeters), while the yield was calculated by weighing the fruits on an electronic balance (FA2104B, Shanghai). After harvest, the root morphology of five cherry radish treated with different doses of wood vinegar was analyzed using the WinRhizo system v.4.0b (Instruments Regent LA2400, Japan). The photosynthetic index was measured on a sunny morning (9:00-11:00), and the second leaf at the top of the plant was selected as the measured leaf (*Gu et al., 2021*). The net photosynthetic rate (Pn), transpiration rate (Tr), stomatal conductance (Gs), and intracellular $CO_2$ concentrations (Ci) of cherry radish were analyzed by the CIRAS-3 portable gas exchange system (PP-Systems, USA) (*Liu et al., 2020*). The supernatant was separated and the absorbance was recorded at 662 nm for Chlorophyll a and 646 nm for Chlorophyll b using the Shimadzu UV-1800 spectrophotometer (Shimadzu, Kyoto, Japan) (*Şükran, Gunes & Sivaci, 1998*).

## Microbial diversity and community

The DNA of soil bacteria was extracted from 0.5 g of frozen soil using the FastDNA SPIN Kit (MP Biomedicals, CA, USA). The 16S rRNA gene was amplified through thermocycler polymerase chain reaction (PCR) system (GeneAmp 9700, ABI, USA) using 16S rRNA amplicon (338F, 5′-ACTCCTACGGGAGGCAGCAG-3′)/806R, 5′-GGAC TACHVGGGTWTCTAAT-3′). Then, the quality and concentration of the extracted DNA were determined using a NanoDrop ND-1000 UV–Vis spectrophotometer (Thermo Fisher, Waltham, MA, USA) and a Quanti Fluor dsDNA system (Promega, Madison, WI, USA). Majorbio Bio-Pharm Technology Co. Ltd. (Shanghai, China) used the Illumina MiSeq platform (Illumina, San Diego, USA) to analyze the bacterial diversity according to the standard protocols. The online Majorbio Cloud Platform (http://www.majorbio.com) was used to perform data analysis, which included Venn diagram analysis, collinearity analysis, alpha diversity analysis, beta diversity analysis, and heatmap analysis.

## Statistical analysis

One-way ANOVA, correlation analysis, and Duncan's multiple range-test were obtained from SPSS version 25.0 (IBM, Armonk, NY, USA). A redundancy analysis (RDA) was conducted utilizing the R programming language to elucidate the interplay between the microbial community structure and the physiological parameters of cherry radish. All the

experiments were carried out in five replicates and a significant difference was considered at $p < 0.05$. Unless otherwise indicated, all other experiments were conducted in five replicates, and significant differences between the experiments were taken into account ($p < 0.05$).

# RESULTS

## Composition analysis of wood vinegar

The molecular composition of wood vinegar from biomass pyrolysis through FT-ICR MS analysis are depicted in Fig. 1. All detected substances are divided into seven groups, which were shown as follows: (lipid (H/C: 1.5–2.0; O/C: 0–0.3), protein (H/C: 1.5–2.2; O/C: 0.3–0.67), carbohydrate (H/C: 1.5–2.2; O/C: 0.67–1.2), unsaturated hydrocarbon (H/C: 0.7–1.5; O/C: 0–0.1), lignin (H/C: 0.7–1.5; O/C: 0.1–0.67), tannin (H/C: 0–1.5; O/C: 0. 67–1.2), and condensed aromatic-like components (H/C: 0.2–0.7; O/C: 0–0.67)), and the results shown in the Van Krevelen diagram clearly illustrated the differences in molecular composition of wood vinegar. The main distribution of wood vinegar compounds $H/C = 0.2 - 0.7$ and $H/C = 0.5 - 1.5$, indicating rich aromatic structure and lignin, with 45.51% and 22.70% contents, respectively (Figs. 1C–1D).

## Wood vinegar promotes the yield of cherry radish

The effects of wood vinegar at different concentrations on the yield indicators of cherry radish are different (Fig. 2). Within a certain concentration range (W50-W200), the yield of cherry radish gradually increased with the increased concentration of wood vinegar. When the added concentration of wood vinegar reached W200, the diameter and yield of cherry radish reached the largest values, increasing by 42.43% and 44.91% respectively. Simultaneously, wood vinegar could also promote the shoot biomass of cherry radish (Fig. S1). After treated by 200-fold diluted wood vinegar, the growth rate and biomass accumulation of cherry radish increased by 34.08% compared with CK. The notable enhancement may be attributed to the regulatory effects of wood vinegar on soil microbial activity, thereby potentially boosting microbial processes that are beneficial for plant growth.

## Effect of wood vinegar on photosynthesis of cherry radish

The effect of wood vinegar treatment on the photosynthesis of cherry radish are shown in Fig. 3. Compared with CK, there was a significant difference in the net photosynthetic rate of cherry radish treated with different concentrations of wood vinegar ($p < 0.05$). After applying 200 times wood vinegar, the maximum net photosynthetic rate reached 25.53 $\mu m$ $CO_2/(m^2 \cdot s)$, which was 80.45% and 19.97–101.84% higher than the CK and other wood vinegar treatment groups, respectively. The net photosynthesis also verified that the W200 treatment group had the strongest ability to accumulate fruit and biomass in cherry radish (Fig. 2). Similarly, 200 times wood vinegar treatment had a positive impact on the transpiration rate of cherries (up to 15.88 mmol $H_2O/(m^2 \cdot s)$), accelerating water absorption and transportation (Fig. 3B). The stomatal conductance fluctuations were significant, with a significant level ($p < 0.05$) observed in the wood vinegar treatment

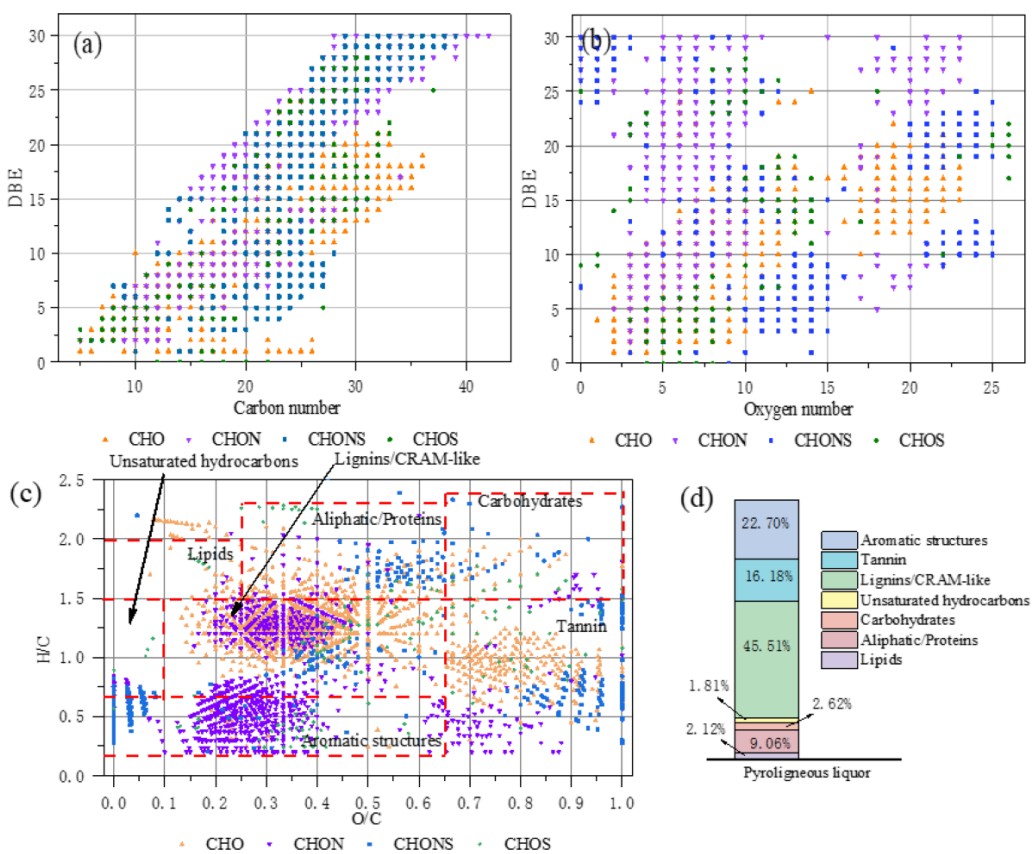

**Figure 1** Plots of carbon (CHO, CHON, CHONS, and CHOS chemicals) (A) and oxygen (CHO, CHON, CHONS, and CHOS chemicals) (B) number *versus* DBE of WV. (C) Van Krevelen diagrams of CHO, CHON, CHONS, and CHOS chemicals for biomass pyrolysis derived-WV, and (D) Bar d.

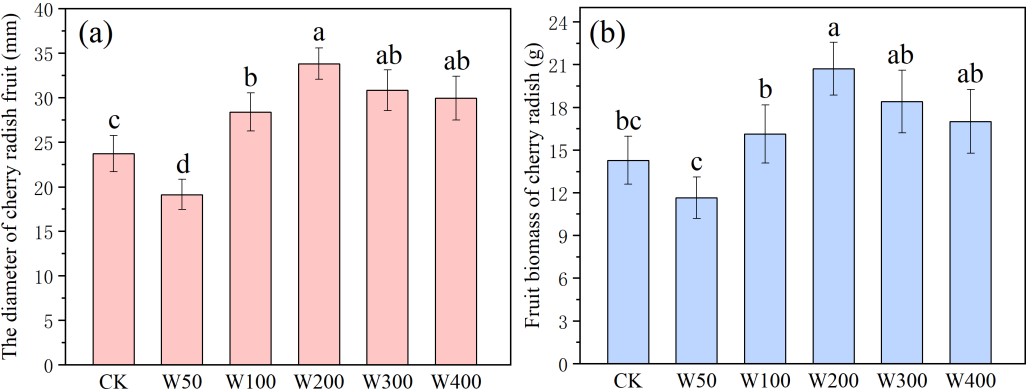

**Figure 2** Effect of wood vinegar treatment on the diameter (A) and weight (B) of cherry radish fruit. Data with different letters are significantly different ($p < 0.05$).

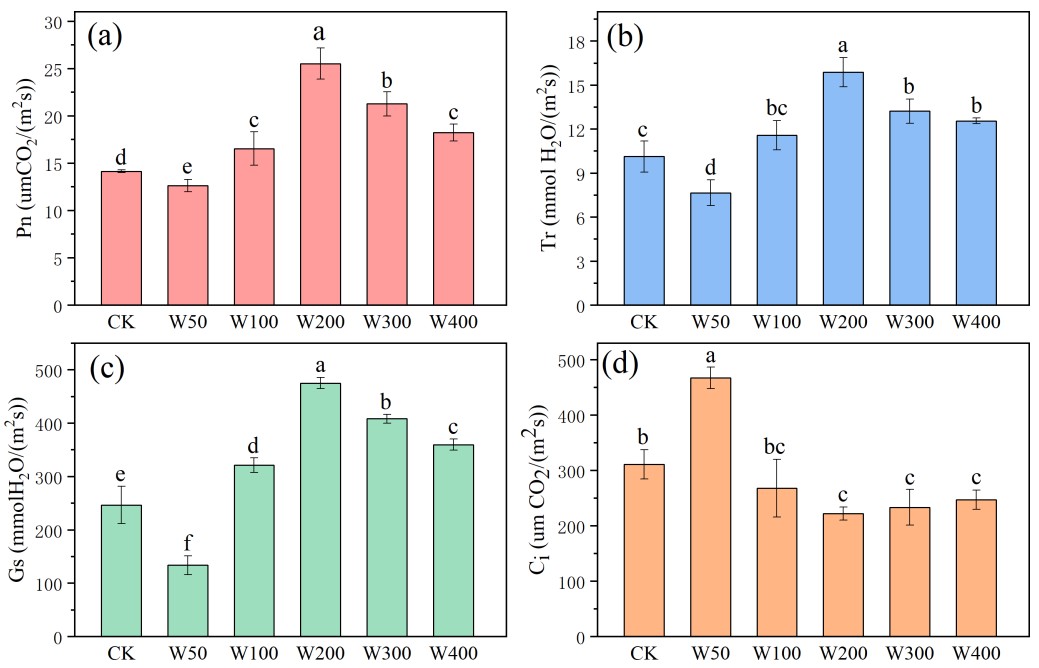

**Figure 3** The gas exchange parameters: (A) net photosynthetic rate (Pn), (B) transpiration rate (Tr), (C) stomatal conductance (Gs), and (D) intercellular $CO_2$ concentration (Ci) of cherry radish.

compared to the control, particularly with W200 showing a 92.57% increase (Fig. 3C). Except for the W50 treatment, the intercellular $CO_2$ concentration in cherry radish leaves of all other treatment groups was lower than that of the CK group, with a decrease range of 13.83%–28.56% (Fig. 3D). However, the lowest intercellular $CO_2$ in W200-treated cherry radish leaves indicated enhanced $CO_2$ utilization during photosynthesis, compared to other treatment groups. The trend of intercellular $CO_2$ content in different treatment groups were basically the opposite of net photosynthetic rate, which was consistent with the laws of plant photosynthesis (*Wang et al., 2011*). In addition, the trends in cherry leaf chlorophyll content and chlorophyll a/b ratio (Fig. S3) were in line with the patterns of changed net photosynthetic rates.

## Effects of wood vinegar on the abundance and diversity of microbial

The coverage of the wood vinegar treatment group (W200 and W400) and the CK were all above 0.99, indicating that the sequencing results well reflected the actual situation of the bacteria and fungi of the samples (Figs. 4A–4B). The Shannon index of the treatment group with 200-fold diluted wood vinegar showed the highest value, indicating that the treatment with 200-fold diluted wood vinegar was beneficial to the increase of rhizosphere bacteria diversity, as shown in Figs. 4A–4B. Simpson's index also showed that the bacterial community of the treatment group with 200-fold diluted wood vinegar had the lowest Simpson index, which further supports the results of the Shannon index. The results showed that W200 had no significant difference in fungal diversity index ($p > 0.05$), but had extremely significant differences in bacterial Shannon ($p < 0.001$), Simpson

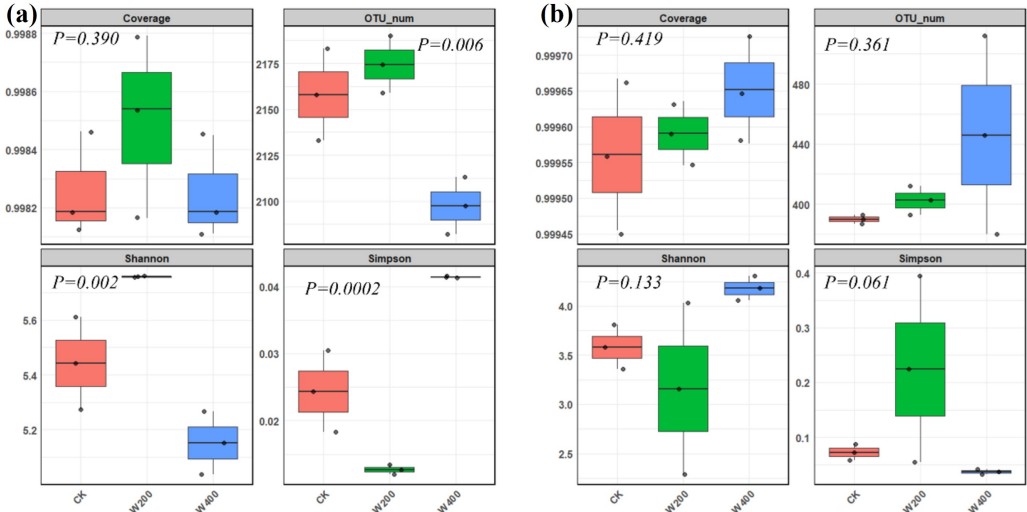

**Figure 4** Effects of wood vinegar on the abundance and diversity of bacteria (A) and fungi (B) in rhizosphere soil.

($p < 0.001$), and OUT number ($p < 0.001$), indicating that W200 can promote bacterial diversity and richness. Venn diagram could be used to represent the similarity and overlap of soil microorganisms (Fig. S4). The operational taxonomic unit (OTU) number of bacteria and fungi in the soil samples after being treated differently were 2047 and 290, respectively. The independent bacteria and fungi OTU numbers in the 200-fold diluted wood vinegar treatment were 148 and 164, accounting for 5.63% and 24.33% of the OTUs, respectively. Results from the data indicated that wood vinegar treatment increased the composition of dominant bacterial and fungal genera in the rhizosphere soil and formed a more diverse microbial environment in the bacterial community structure, thereby benefiting the growth of cherry radish.

## DISCUSSION

### Cherry radish yield and photosynthesis improvement under wood vinegar application

Results indicated a positive effect of wood vinegar application on biomass production of cherry radish, which might be related to the regulation of wood vinegar on soil microbial activity, and possible by promoting cherry radish nutrient absorption. *Akley et al. (2023)* recorded that the soil drenching of wood vinegar enhanced soil enzyme activity (arylsulphatase, acidic phosphatase, and α-Glucosidase), which strongly correlates with improved biomass and grain yield of cowpea. Similarly, *Lu et al. (2019b)* discovered that using low concentrations of wood vinegar (0.33−0.50 mL/L) stimulated root vitality and enhanced fresh root biomass of wheat. Moreover, the application of wood vinegar increased soil organic matter, for wood vinegar stimulated more labile C pool which enhanced uptake of nutrients by plants, and stimulated populations of beneficial microorganisms (*Cardelli et al., 2020*; *Wang et al., 2022*). Soil treatment with a 500-fold diluted wood vinegar resulted

in a significant ($p < 0.05$) 73.39% increase in bacterial population (*e.g.*, Gram-negative bacteria, anaerobic bacteria, and aerobic bacteria) and enhanced soil ecological activity (*Rui et al., 2014*). Overall, the use of wood vinegar could improve the growing environment of cherry radish, thereby promoting aboveground growth and development of cherry radish (*Mhamdi, 2023*).

Compared with the CK and other wood vinegar concentrations, the cherry radish treated with 200-fold diluted wood vinegar showed significant differences ($p < 0.05$) in net photosynthetic rate, transpiration rate, stomatal conductance, and intercellular $CO_2$ concentration. After soil drenching with 200-fold diluted wood vinegar, the net photosynthetic rate and transpiration rate of cherry radish were 19.97–101.84% and 20.03–107.16% higher than the CK and other wood vinegar treatment groups, respectively. It was reported that wood vinegar enhanced net photosynthesis by regulating soil microbial activity and enhancing the ability of crop roots to absorb nutrients (*Chen et al., 2016*; *Rui et al., 2014*). The transpiration rate of cherries was related with root growth (*Kulkarni et al., 2017*), and the changed trend of root index in W200 treated cherry radish further confirmed the enhanced transpiration rates (Fig. S2). Cherry radish transpiration was vital for preserving water balance and facilitating photosynthesis, while robust root growth ensured efficient water and nutrient uptake from the soil (*Farooq et al., 2019*; *Wu et al., 2022*). For example, after W200 treatment, the contents of root tips, length, and surface area were 35.50, 16.70 cm, and 36.33 cm$^2$, respectively, with an increase of 59.19%, 74.14%, and 26.72% compared to CK. As the concentration of wood vinegar increased, the stomatal conductance of cherry radish firstly decreased and then increased, which was consistent with the changes in net photosynthetic rates and transpiration rates. The trend of change may be attributed to the high concentration of wood vinegar that is impeding the growth of cherry radish by passivating soil microbial activity. The results showed that cherry radish treated with 200-fold wood vinegar exhibited a reduction in intercellular $CO_2$ concentration ranging from 11.33 to 245.50 $\mu$m $CO_2/(m^2s)$ compared to other treatment groups. This indicated that the leaves could utilize more $CO_2$ for photosynthesis (*Gu et al., 2021*). The overall trend of chlorophyll content in cherry radish was consistent with the net photosynthetic rate of the leaves, with chlorophyll a and chlorophyll b increasing by 24.93–120.22% and 11.01–89.73% compared to the treatment group, respectively. Similarly, the exposure of compost to a 0.5% wood vinegar treatment resulted in a marked elevation of 21% in the chlorophyll a/b ratio among *cucumis sativus* seedling compared to those untreated with wood vinegar (*Afsharipour, Mirzaalian & Seyedi, 2024*). The increase in chlorophyll content could effectively promote photosynthesis by enhancing the plant's ability to absorb light energy, leading to improved growth, higher biomass production, and better overall plant health (*Li et al., 2020*). In a word, the application of a 200-fold diluted wood vinegar significantly improved the photosynthetic rate, transpiration, root growth, and chlorophyll content in cherry radish, thereby enhancing nutrient absorption and effective utilization of intercellular $CO_2$.

## Wood vinegar enhances rhizosphere microbial activity

After wood vinegar treatment, there were significant differences in the relative abundance of bacterial communities at both the phylum and genus levels (Fig. 5). After 200-fold diluted wood vinegar treatment, the proportion of *Actinobacteriota* and *Firmicutes* in soil samples increased, with respective increases of 40.88% and 126.67% compared to the CK group. *Actinobacteriota* could promote the decomposition of soil organic matter and nutrient transformation, thereby contributing to the improvement of soil fertility (*Lan et al., 2022*; *Ren et al., 2018*). The wood vinegar compounds contained 68.21% organic matter, which stimulated the activity of actinomyces and thereby increased its abundance. In addition, *Actinobacteriota* also had a certain bioprotective effect, thus inhibiting the growth of pathogenic bacteria and promoting the healthy growth of cherry radish. Meanwhile, the abundance of *Proteobacteria* (up to 25.82%) in high carbon soil was promoted by W200 application (*Fierer, Bradford & Jackson, 2007*), and it was considered as an important soil symbiont together with *Firmicutes* (*Anderson et al., 2018*; *Zhalnina et al., 2015*). Moreover, the bacterial genera with relative abundance greater than 1% in soil treated with wood vinegar at different dilution concentrations and CK were *Arthrobacter*, *Sphingomonas*, *Vicinamibacterales*, *Vicinamibacteraceae*, *Massilia*, *g__RB41*, *norank_f__norank_o__norank_c__KD4-96*, and *norank_f__Roseiflexaceae* (Fig. 5B). More importantly, the abundance of *Arthrobacter*, *Sphingomonas*, and *norank_f__ norank_o__Cyanobacteriales* genera in wood vinegar at a concentration of 200 times was significantly higher (over 10%) than other treatments, indicating a significant change in the proportion of dominant bacterial genera in the rhizosphere soil of cherry radish. Especially for *norank_f__ norank_o__Cyanobacteriales*, which were between 15.34 and 230.05 times higher than CK and W400. Notably, wood vinegar additives stimulate cherry radish to absorb rhizosphere elements, degrade variety aromatic compounds (the main component of wood vinegar), and drive nitrogen cycling by increasing the abundance of dominant soil microorganisms, such as *Arthrobacter* (*Schwabe et al., 2021*), *Sphingomonas* (*Gong et al., 2016*; *Liu et al., 2017*), and *norank_f__ norank_o__Cyanobacteriales* (*Wang et al., 2023*).

Additionally, compared to CK and W400 treatments, the fungi in the W200 treatment group showed a significant increase (0.98–25.06 fold higher) in the main phylum, specifically *Basidiomycota*, *unclassified_k__Fungi*, and *Blastocladiomycota*, with the main fungal genera being *unclassified_f__Stachybotryaceae*. *Basidiomycota*, as a crucial decomposer, generates enzymes (peroxide) that enzymatically break down plant components such as cellulose and lignin, thereby augmenting the overall carbon pool of the soil (*Ahmed, De Figueroa & Pajot, 2020*). Moreover, *unclassified_k__Fungi*, as a dominant genus in rhizosphere soil, stimulates fungi that generate total nitrogen (*Yang et al., 2022*). The organic matter (W200) in the rhizosphere promoted the abundance of *Blastocladiomycota* (*Zhang et al., 2021*), which was beneficial for crop growth. Similarly, *Stachybotryaceae* also showed a positive response to nitrogen in the rhizosphere soil (*Kumar et al., 2022*). These results also indicated that fungal communities displayed a vital role in the utilization and absorption of soil nutrients by crops.

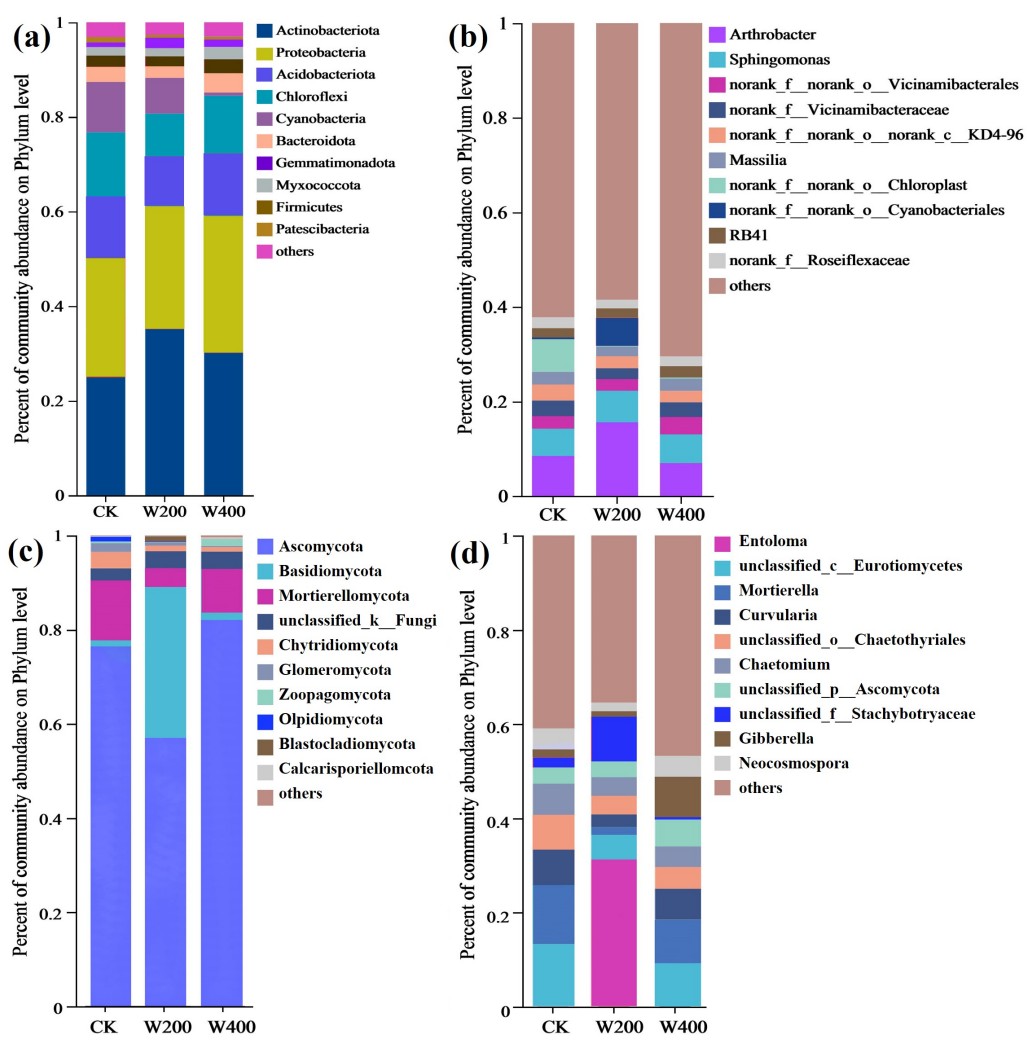

**Figure 5** The relative abundance of soil bacteria (phylum (A) and genus (B)) and fungi (phylum (C) and genus (D)).

## The relationship between cherry radish growth factors and microbial communities

RDA was employed to explore the relationship between environmental factors and microbial communities during the application of wood vinegar as fertilizer (Figs. S5–S6). RDA1 and RDA2 accounted for 65.89% and 23.95% of the variance in bacterial community dynamics, and 39.99% and 28.07% for fungal communities, respectively. During the cultivation process, the order of correlation between environmental factors and bacterial/fungal communities was: surface area >chlorophyll b >shoot biomass >length >fruit biomass for bacteria, and Tr >surface area >Pn >Gs >fruit biomass for fungi. The strong correlations ($p < 0.05$) indicated that optimizing environmental parameters could regulate microbial communities, enhancing cherry radish growth. Moreover, the correlation plot generated in Fig. 6 and Fig. S7 identified

the microbial communities and growth factors mediating the promotion of cherry radish growth by wood vinegar. The root growth indicators (length, tip, surface area, and diameter) showed significant correlation ($p < 0.05$) with the genera of *Arthrobacter*, *norank_f__norank_o__Cyanobacteriales*, and *Agromyces* (Fig. S7). Among them, the *Arthrobacter* and *norank_f__norank_o__Cyanobacteriales* in the W200 treatment group were 1.85–297.85 times higher than those in the CK and W400 treatment groups. It has been reported that *Arthrobacter* (*Schwabe et al., 2021*) and *norank_f__norank_o__Cyanobacteriales* (*Wang et al., 2023*) degrade aromatic compounds (wood vinegar) to drive the cycling of nitrogen nutrients in the rhizosphere. Meanwhile, *norank_f__norank_o__Cyanobacteriales*, *Ellin6067*, and *Agronomyces* exhibit a significant relationship ($p < 0.05$) with photosynthesis. In the soil treated with W200, it was found that *norank_f__norank_o__Cyanobacteriales* (*Wang et al., 2023*) and *Agronomyces* (*Wang et al., 2020*) were involved in nitrogen and carbon cycling, respectively, thereby promoting cherry radish growth and significantly improving physiological indicators.

For fungal communities, *Allomyces*, *Clonostachys*, *Fusarium*, *Fusicolla*, *Knufia*, *Nigrospora*, *Paraconiothyrium*, *Preussia*, *Talaromyces*, *Trichocladium*, *unclassified_f__Didymellaceae*, *unclassified_f__Stachybotryaceae*, *unclassified_Mortierellomycota* showed a positive response ($p < 0.05$) to root length, root number, and root surface area of cherry radish, while also showed a positive correlation ($p < 0.05$) with aboveground biomass and fruit (Fig. 6). The fungi that positively correlated ($p < 0.05$) with the photosynthesis (except for Ci) of cherry radish include *Albifimbria*, *Allomyces*, *Calcarisporiella*, *Clonostachys*, *Fusarium*, *Fusicolla*, *Knufia*, *Nigrospora*, *Paraconiothyrium*, *Preussia*, *Talaromyces*, *unclassified_c__Sordariomycetes*, *unclassified_Chytridiomycota*, *unclassified_f__Didymellaceae*, *unclassified_f__Stachybotryaceae*, and *unclassified_Mortierellomycota*. Particularly, *Allomyces*, *Fusarium*, *Fusicolla*, *Trichocladium*, and *unclassified_f__Didymellaceae*, showed a very significant correlation with cherry radish growth and physiological indicators ($p < 0.01$). The common bacterial species (such as, *Cladorrhinum* (*Barrera et al., 2019*), *Fusarium* (*Yuan et al., 2020*), *Allomyces* (*Zhang, Huang & Liu, 2013*), and *Clonostachys* (*Fournier et al., 2020*) in the soil could decompose the organic matter (wood vinegar), or wood vinegar could activate these bacteria to enhance the absorption of nutrients by cherry radish, thus promoting the development of its root system and ultimately boosting crop photosynthesis and yield. Studies have already verified that *Fusarium* and *Fusarium* were the main rhizosphere populations that promoted rapeseed growth and resulted in a higher yield (*Lay et al., 2018*). Data showed that wood vinegar stimulated the dominant microorganisms in the cherry radish rhizosphere, thus increasing nutrient absorption and boosting yield.

## CONCLUSIONS

In this study, the effects of wood vinegar on physiological indicators and rhizosphere microbial community of cherry radish were studied through soil culture experiments. The results indicated that within a certain concentration range of W50 to W200, cherry radish yield gradually increased with rising wood vinegar concentration. Specifically, at

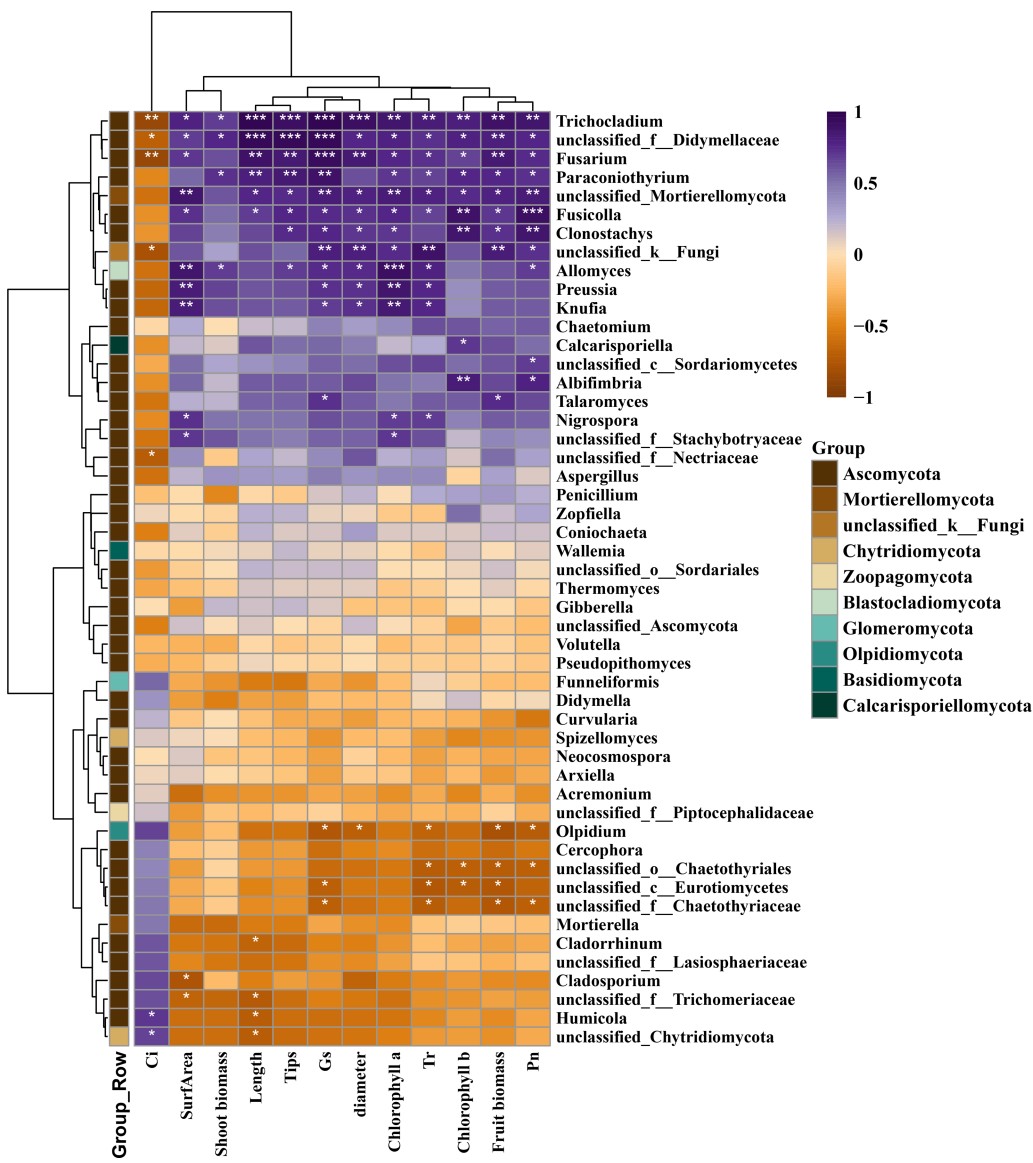

**Figure 6** **Thermogram analysis of the correlation between fungi community and growth factors.** Note: * Significant correlation at 0.05 level, ** Very significant correlation at 0.01 level. *** Very significant correlation at 0.001 level.

an application concentration of W200, the diameter, biomass, and yield of cherry radish increased by 42.43%, 34.08%, and 44.91%, respectively. Furthermore, after applying wood vinegar (W200), the maximum net photosynthetic rate of cherry radish leaves reached 25.53 μm $CO_2$/(m² · s), exhibiting a similar growth trend as other photosynthetic indicators. Although the optimal range for performance improvement has been established, dilutions exceeding this range, such as W300 and W400, also exhibited better performance than CK. In particular, for certain indicators, these higher dilutions (W300 and W400) showed similar levels of improvement as the W200 concentration. The application of wood

vinegar has been proven benefit the growth and reproduction of microbial communities. Specific microorganisms, such as *Allomyces*, *Fusarium*, and *Fusicolla*, have been shown to significantly enhance the growth of cherry radish ($p < 0.01$). The promoting effect may be related to the increased soil microbial activity or enhanced nutrient uptake facilitated by wood vinegar. Due to diverse range of biomass types and pyrolysis temperatures, further research is needed to understand the effects of wood vinegar on the growth and yield of cherry radish, as well as its potential benefits for other plants.

### Funding
This work was supported by the Key Research Project of Natural Science in Colleges and Universities of Anhui Province (2022AH051096), the Funding Project for Young Backbone Teachers to Visit and Study in China (JNFX2023061), the Chuzhou University Research Initiation Fund Project (2023qd49), the National Natural Science Foundation of China (42207435), the Natural Science Foundation of Hebei Province, China (B2023202077), and the China Postdoctoral Science Foundation (2020M680868). The funders had no role in study design, data collection and analysis, decision to publish, or preparation of the manuscript.

### Grant Disclosures
The following grant information was disclosed by the authors:
The Key Research Project of Natural Science in Colleges and Universities of Anhui Province: 2022AH051096.
Funding Project for Young Backbone Teachers to Visit and Study in China: JNFX2023061.
Chuzhou University Research Initiation Fund Project: 2023qd49.
National Natural Science Foundation of China: 42207435.
Natural Science Foundation of Hebei Province, China: B2023202077.
China Postdoctoral Science Foundation: 2020M680868.

### Competing Interests
The authors declare there are no competing interests.

### Author Contributions
- Shiguo Gu conceived and designed the experiments, analyzed the data, prepared figures and/or tables, authored or reviewed drafts of the article, and approved the final draft.
- Wei Zhu conceived and designed the experiments, analyzed the data, authored or reviewed drafts of the article, and approved the final draft.
- Liying Ren performed the experiments, analyzed the data, prepared figures and/or tables, and approved the final draft.
- Binbin Sun conceived and designed the experiments, analyzed the data, prepared figures and/or tables, authored or reviewed drafts of the article, and approved the final draft.
- Yuying Ren performed the experiments, authored or reviewed drafts of the article, and approved the final draft.

- Yongkang Niu performed the experiments, authored or reviewed drafts of the article, and approved the final draft.
- Xiaokang Li performed the experiments, authored or reviewed drafts of the article, and approved the final draft.
- Qingshan He conceived and designed the experiments, analyzed the data, prepared figures and/or tables, and approved the final draft.

## DNA Deposition

The following information was supplied regarding the deposition of DNA sequences:

The 16S rRNA gene was amplified through thermocycler polymerase chain reaction (PCR) system (GeneAmp 9700, ABI, USA) using 16S rRNA amplicon (338F, 5′-ACTCCTACGGGAGGCAGCAG-3′)/806R, 5′-GGAC TACHVGGGTWTCTAAT-3′).

## Data Availability

The raw sequence reads are available at NCBI: PRJNA1120623.

https://www.ncbi.nlm.nih.gov/bioproject/PRJNA1120623/

## Supplemental Information

Supplemental information for this article can be found online at http://dx.doi.org/10.7717/peerj.18505#supplemental-information.

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
