# Peer review of "New insights into the impact of wood vinegar on the growth and rhizosphere microorganisms of cherry radish (*Raphanus sativus* L.)"

_PeerJ, doi:10.7717/peerj.18505_

## Round 0.1 · original submission · Major Revisions

Authors should also note the queries raised by reviewer 4 in their attached PDF

·

Basic reporting

The English should be revised and the article needs to conform standard of courtesy and expression

Experimental design

Need revision and more details

Validity of the findings

Conclusions are not well stated

Additional comments

The findings are innovative but the manuscript needs revision and review by English speaking professional

Reviewer 2 ·

Basic reporting

The manuscripts provide information about the use of wood vinegar to enhance radish growth by increasing microbial activity. The paper presents the proposed benefits and describes the materials and methods used appropriately. While the results support the hypothesis, the manuscripts lack a clear discussion. This can be improved by adding references and providing a more detailed explanation.

Experimental design

The investigation is conducted rigorously, adhering to high technical and ethical standards. The methods are described in sufficient detail to allow for replication, to ensure the reliability and validity of the findings.

Validity of the findings

The results are fine. However, the manuscript lacks a clear discussion section. To improve, it is recommended to add references and provide a more detailed explanation of the results and their implications. This will strengthen the overall contribution of the research.

Additional comments

However, the manuscript lacks a clear discussion section. To improve, it is recommended to add references and provide a more detailed explanation of the results and their implications. This will strengthen the overall contribution of the research.

Annotated reviews are not available for download in order to protect the identity of reviewers who chose to remain anonymous.

Reviewer 3 ·

Basic reporting

Correct the botanical name as per the rule in the title. The first letter is the capital of the genus and species in lowercase. This is a very silly mistake. Try to correct it in the whole MS.
Arrange the keywords alphabetically, and they should be different from the title.
Add the latest references in the introduction section. References are ancient
NaClO write it as sodium hypochlorite, then (NaClO)
Stage and How Much Product you have analyzed (number of plants per replication)or number of fruit per treatment
Figure should be described in sequence (one figure 2c and 2d u have described in front then later on u have described fig 2)
The result and discussion need to be rewritten, and add latest references in MS
Some typographical errors are present in MS
Follow the journal guidelines strictly

Experimental design

Research question well defined, relevant & meaningful. It is stated how research fills an identified knowledge gap

Validity of the findings

Correct the botanical name as per the rule in the title. The first letter is the capital of the genus and species in lowercase. This is a very silly mistake. Try to correct it in the whole MS.
Arrange the keywords alphabetically, and they should be different from the title.
Add the latest references in the introduction section. References are ancient
NaClO write it as sodium hypochlorite, then (NaClO)
Stage and How Much Product you have analyzed (number of plants per replication)or number of fruit per treatment
Figure should be described in sequence (one figure 2c and 2d u have described in front then later on u have described fig 2)
The result and discussion need to be rewritten, and add latest references in MS
Some typographical errors are present in MS
Follow the journal guidelines strictly

Additional comments

Correct the botanical name as per the rule in the title. The first letter is the capital of the genus and species in lowercase. This is a very silly mistake. Try to correct it in the whole MS.
Arrange the keywords alphabetically, and they should be different from the title.
Add the latest references in the introduction section. References are ancient
NaClO write it as sodium hypochlorite, then (NaClO)
Stage and How Much Product you have analyzed (number of plants per replication)or number of fruit per treatment
Figure should be described in sequence (one figure 2c and 2d u have described in front then later on u have described fig 2)
The result and discussion need to be rewritten, and add latest references in MS
Some typographical errors are present in MS
Follow the journal guidelines strictly

·

Basic reporting

The article was written in a clear language and was concise. However, Authors need to revised some portion of the abstract. Information was plant parameters was not included in the abstract. For instance:
Line 16-17: Authors should revise this sentence to “use in crop production and environmental safety”
Line 22-23: Authors should indicate percent (%) increased of the abundance of Actionbacteriota and Fimicutes
Line 28-29: The authors should include information on plant parameters that were influenced by the Wood vinegar. It appears that the authors focused on more the vinegar effect on soil microbial population.
Line 31-32: Authors should kindly provide the information on the exact concentration of wood vinegar or dilution of wood vinegar that promote cherry plant growth and rhizosphere microbiome

The introduction was well written and supported by recent citations that are within the last 10 years. The research background was well provided and research objective clearly stated. The research gap was not well exploit and I recommend further work for the research gaps to be explicitly stated. The research hypothesis was stated but requires some modification.

I also recommend some revision to be made to the introduction /background

Line 50-52: The authors should double check this statement. Several authors have documented Wood vinegar effect on soil microbiome. examples Jeong et al :2015: Effects of Rhizosphere Microorganisms and Wood Vinegar Mixtures on Rice Growth and Soil Properties

Line 53: Spell WV in full "wood vinegar"
Line 56: Remove one of the bracket "(K)
Line 60-62: Authors should kindly reword this sentence as "Poultry manure compost fortified with WV signficcantly increase wheat yield through WV leaching solluble salts and enhancing the effect uptake of P and K"
Line 70-71: In this study we used wood vinegar produced from pyrolysis of rice straw.
Line 74-75: Authors should kindly rewrite the research hypothesis

In general the authors did a good job, NO serious English language editing is required but I recommend the authors give to the manuscript to a native English speaker to read through again.

Experimental design

This research falls within the scope of the journal. Authors need to provide more information on the methodology used. For instance Authors need to provide more information on the brown soil ; thus, they should provide information on soil taxonomy and cropping system as well as the ecology of the areas where the spoil samples were collected. The author should provide more information on the on how the rice straw was prepared into Wood vinegar?

The research design used was adequate and suitable to achieve the research question being addressed.
Methodology used was rigorous and meet a technical standard. The number replication was ok, thus five replications, and the treatments were clearly indicated. Microbial diversity and community was assessed using the appropriate methodology which is fine.

However, some modifications are required. Line 90-91: need to be rewritten for clarity. Line 93-96: The authors should provide more information on how the dilution was done and was water tap water or distilled water? Line 114-115: Authors should provide more information on how the yield was calculated?

Validity of the findings

There is no new novelty here except that the authors evaluated the impact of Wood vinegar on cherry raddish. Presently, there is no wood vinegar study which targeted cherry raddish. Hence, this study can potentially boost cherry raddish production value chain thereby helping to ensure food security. This is because the authors established that the wood vinegar dilution of 200 time can improve cherry raddish.

The data analysis was ok and appropriate for this work. Authors also used correlation analysis to established relationship between the crop parameters with the microbiome data before drawing some key conclusions. However. I would recommend the use of multivariate analysis such as principal component analysis, to help establish the key relationship between the parameters before some key conclusions are made from this work.

In additions, the authors needs to make some modifications to their results. For instance
Line 165-166" Authors should explain how WV can increase SOM in the short-term?
I think, WV stimulated more labile C pool which ehanced uptake of nutrients by plants.

Line 170-173: Authors need explain why some of the other treatments (dilution) that oupterformed the CK were not clearly captured in the write -up.
Line 180-182: This is partly true statement. Authors should revisit and graph and offer more explanation
Line 185-189: Authors need to explain further and give reasons for this observations

Line 189-191: What about in relation to the other indicators?

Line 192-194: Among the WV dilution, Stomatal conductivity declined after 200 but SC at 300 and 400 were higher than 50 and 100? Authors should explain what accounted for this variability?

Line 195-197: Please was the comparison in relation to the control? what about the other treatment ?

Line 199-202: Provide citation to back their assertions.

Line 199-202: Author should also explain the other WV treatment?

The conclusion was appropriate but all the data was compared to the control (CK). There were some treatments that performed similarly to the CK but were not captured. Therefore, the conclusion needs some modification.
Line 296-301: The authors should also make inference on some promising dilution. For instance Some of the out dilution (300 and 400) also outperformed the control and also yielded similar output the 200; and need to be discussed .

Line 306- 308: Authors should revise this statement. It is not clear.

---

## Round 0.2 · accepted · Accept

The authors have addressed all of the reviewers' comments, this manuscript is ready for publication

Reviewer 2 ·

Basic reporting

the authors have been revised all suggestion.

Experimental design

appropriated research design

Validity of the findings

clearly stated

Reviewer 3 ·

Basic reporting

Authors addressed all the said comments

Experimental design

fine

Validity of the findings

fine